# Mind Over Matter: Mindfulness, Income, Resilience, and Life Quality of Vocational High School Students in China

**DOI:** 10.3390/ijerph17165701

**Published:** 2020-08-07

**Authors:** Shannon Cheung, Xiaoxia Xie, Chien-chung Huang

**Affiliations:** 1School of Social Work, Rutgers University, New Brunswick, NJ 08901, USA; scheung@ssw.rutgers.edu; 2Research Institute of Social Development, Southwestern University of Finance & Economics, Chengdu 611300, China

**Keywords:** income, mindfulness, resilience, quality of life, vocational education, high school students, China

## Abstract

Many social welfare programs focus on the provision of cash assistance and cash transfers to improve the quality of life (QoL) of those living in low-income households. While there is literature to support a positive relationship between income and QoL, studies have shown that QoL is impacted by non-income-related factors. This study examined the effects of income and mindfulness on QoL through a mediator, resilience, and attempts to answer the question of how important income is to QoL, relative to a non-income-based determinant, mindfulness. Using a sample of 905 emerging adults from the senior class of a secondary vocational high school based in an impoverished county of China, we studied two key determinants of QoL, income and mindfulness, as well as respective pathways, during a particularly critical stage of life. The results indicated that mindfulness had strong direct and indirect effects on QoL via resilience, while income had only limited indirect effects on QoL via resilience. Policy implications were discussed.

## 1. Introduction

The concept of quality of life (QoL) refers to an individual’s perception of their ability to meet goals and standards within the facets of education, health and well-being, and financial development and in the context of their culture and value system [1,2,3]. Health-related QoL research has been studied across the life course within the context of medical conditions [4,5] as well as different socioeconomic and health factors [6,7,8,9,10,11]. Many other studies have discussed how material capacity [12,13,14] and mental capacity [15,16] each affect QoL, but only a few studies have inquired how material capacity and mental capacity may differentially affect QoL. The question of whether one has greater influence on QoL than the other poses necessary guidance for the future of social services. Thus far, income-based social programs have dominated the social welfare landscape; however, increased income may not necessarily lead to improved QoL due to lack of access to non-income-related determinants [17]. This calls for further research on the potential of non-income-based social programs and methods of improving QoL. The aim of this study was to examine the effects of material and mental capacity on QoL in a sample of senior vocational students in China—a group of individuals entering a critical period of development as they transition from adolescence into emerging adulthood.

## 2. Literature Review

### 2.1. Quality of Life and Material Capacity

QoL, which encompasses an individual’s functioning in domains such as physical, emotional, academic, occupational, and social, among low-income populations has been of scholarly, government, and corporate interest recently [18,19,20,21,22]. In past decades, China has doubled down on its efforts to eradicate poverty among its most impoverished populations, an important initiative in the realm of improving QoL, considering the existing cross-cultural evidence of economic determinants of QoL. Socioeconomic status (SES) indicators, including income, were found to predict physical and psychological well-being, bullying, and perceived financial resources of approximately 1150 adolescents, aged 12–18 years, from 7 countries in Europe [14]. In a nationwide sample of over 23,000 Iranian students, aged 6–18 years old, SES was found to have positive associations with school functioning, psychosocial functioning, and health-related QoL as measured using a Persian version of the Pediatric Quality of Life Inventory [12]. Geyer and Peter (2000) found that among three SES indicators, the mortality-related effects of income override those of the other two. Thus, income has a particularly strong association with health-related inequalities [13]. In another study, family income was found to be associated with QoL in patients with chronic kidney disease [20]. Lemos et al. (2015) found that although gender and age influenced QoL, family income was the most important factor affecting QoL in a sample of 170 individuals [20]. Similarly, Ackerman and Paolucci (1983) found, using data collected from a nationally representative sample of over 1000 adults, that both objective and subjective income adequacy had positive relationships with three different life quality measures [23].

Currently, there is evidence to suggest that stable employment and adequate income act as protective factors that contribute to overall family resiliency by reducing financial stress and improving family functioning [24]. By contrast, those of lower socioeconomic status in a community sample of 760 respondents had lower resilience to stress [25], indicating that income can adversely affect children and adults of low-income families. Without adequate employment supports, physical and mental health problems are exacerbated among low-income mothers [26]. Maternal employment has thus been cited to protect low-income mothers from developing mental illnesses such as major depressive disorder [26]. In consideration of the literature that has supported a positive relationship between resilience and QoL [27,28], it is clear that those living in poverty are at a severe disadvantage and at risk of functioning poorly in many domains of life. Given the existing research that connects income, resilience, and QoL together, it is prudent that in recent decades, China has had a marked interest in devising anti-poverty policies and programs to assist rural low-income populations [22,29]. Scholars have researched China’s Minimal Living Standard Guarantee (*dibao*) program and its effects on impoverished urban adults [30] and rural adults [31]. Recently, some studies have begun to examine how cash assistance programs and family income affect younger populations and their financial well-being [32].

### 2.2. QoL and Mental and Emotional Capacity

In addition to material capacity, literature has pointed to non-financial determinants of QoL [18]. In Mayer (1997), it was found that money, per se, does not necessarily have strong effects on child outcomes when considering other life factors [33]. Indeed, literature on subjective well-being and QoL has found that both can be affected by psychosocial factors. For example, cognitive appraisal processes, the way that individuals think about or perceive their life circumstances, have been found to be associated with QoL [16]. This was mediated by resilience; people with greater resilience focused on developing their relationships, pursuing dreams, fostering independence, and having a balanced lifestyle, rather than anticipating their declining health [16]. Schwartz et al. (2017) noted that coping can lead individuals to change the way that they perceive stressors, leading to improved resilience and, in turn, higher QoL [16]. Mindfulness may be a key accessible coping tool to improve resilience and QoL. Bajaj and Pande (2016) found that mindfulness predicted life satisfaction and affect, both indicators of subjective well-being [15]. In another study that used a sample of over 300 Chinese adults, mindfulness predicted life satisfaction [34]. Trait mindfulness has been associated with greater psychological adjustment following trauma exposure [35] by redirecting appraisal and buffering effects of perceived stress [36]. This implicates experiential avoidance in the etiology and maintenance of mental illnesses like depression, anxiety, and post-traumatic stress disorder. When individuals actively tend to day-to-day life without judging their experiences, thoughts, or actions, they are more able to healthily confront personal challenges and stressors. In other words, they demonstrate resilience, a trait that helps individuals cope with adversity and “bounce back” from the impact of adversity [37].

Several studies have found positive relationships between mindfulness and resilience [15,38,39,40]. Meanwhile, resilience and life satisfaction were shown to have a significant positive relationship in a sample of 1400 Chinese college students [41]. Given the relationship between mindfulness and resilience, as well as the relationship between resilience and overall well-being, researchers have begun to examine whether mindfulness has an indirect effect on life quality through resilience. In particular, Bajaj and Pande (2016) found that resilience mediates the relationships between mindfulness and well-being, supporting the role of resilience in the mechanism for the relationship between mindfulness and QoL [15].

### 2.3. Material and Mental Capacity and QoL in Emerging Adults

Although research has shown that both material capacity and mental capacity improve QoL through resilience, most of the existing research has relied on samples of young children, adults, and seniors. For studies that have focused on children, samples covered a wide range of individuals at very different stages of life. The results of such studies can eclipse the experiences of young adults who are at the cusp of emerging adulthood. The years immediately following high school education, ages 18–25, are a unique phase of life, characterized by the transition from dependence on parents and guardians to self-sufficiency across several life domains, including finances [42,43]. Emerging adulthood, according to Blomquist (2007), is “important when looking at realistic outcomes for employment and living independently for all youth” (p. 297) [44]. For seniors at vocational secondary schools in China, this is especially salient considering that they are nearing the end of the education and training needed to provide the foundation for their incomes and occupational achievements for the rest of their working lives. These students are on the brink of entering the working world. Beyond this, emerging adulthood is generally described as “the period of life that offers the most opportunity for identity exploration in the areas of love, work, and worldviews” (Arnett, 2000, p. 473) [42]. Despite the distinct nature of this stage of life, less is known about how income and mindfulness affect QoL among emerging adults.

Beyond this limitation of existing QoL studies, most QoL studies stop short of indicating the difference in effect sizes of material capacity and mental capacity. For example, Friedland et al. (1996) reported that both income and perception-oriented coping were positively associated with QoL, but did not discuss any differences between each factor’s relationship with QoL [7]. The question of whether material or mental capacity is more influential has important implications for the way that publicly funded programs and social service professionals approach the problem of QoL in the general population. This question is also timely, considering recent political platforms based on the implementation of universal basic incomes (UBIs). For example, the former U.S. presidential candidate Andrew Yang ran his campaign based on his signature policy, the “Freedom Dividend”, which would provide a UBI of $1000 a month to every American adult [45]. Yang asserted that his UBI policy could be “a foundation on which a stable, prosperous, and just society can be built” [45], though Mayer’s [33] review has shown that income is not the only determinant of QoL, life achievements, and/or life satisfaction.

Given the above relationships among mindfulness, income, resilience, and QoL, in this study, we aimed to answer the following question: what are the relationships among income, mindfulness, resilience, and quality of life of Chinese vocational students? In this paper, we examined whether income or mindfulness had more dominant effects on the QoL senior students who were on the verge of entering the job market at a vocational school in Southwest China.

## 3. Data and Method

### 3.1. Data and Sample

The data for the present study came from senior students at a vocational school in Gulin County of Sichuan Province, China (hereafter referred to as “School G”). There was a total of 1004 students in the senior cohort at School G, and 930 students were available at the time of survey. A self-administered survey, distributed in classrooms, was conducted in December 2019. After deleting cases with incomplete data, the final analytic sample size was 905 (97% of *N*). School G is the only vocational school in Gulin County, which has a high poverty rate. Since income was one our key variables, we aimed to understand the effects of income, along with other key variables, on QoL for students in high poverty areas. Secondary vocational education in China is conducted at the high school level and includes a variety of vocational training. In particular, School G offers 9 majors to its students, including Computing, Accounting, Tourism, Kindergarten Education, Clothing, Machinery, Auto Repair, Electronics, and Aviation. About 55% of students were male. The average age of the sample was 18 years, and 96% of the sample were Han Chinese. Admission to School G is based on applicants’ scores from the standardized entrance exam used for admittance to all high schools nationwide. Although the maximum score for this exam is 760, School G admits students with minimum scores of 280. By comparison, other high schools only admit students with scores of 470 and above. The research protocol was approved by the research review committee at one of the co-authors’ university in China. An informed consent process was implemented prior to the survey. Students were informed of their voluntary participation as well as their ability to discontinue survey participation at any time.

### 3.2. Measures

The outcome variable was QoL, measured by the Pediatric Quality of Life Inventory (PedsQL), version 4.0, child report. The PedsQL has been shown to have high reliability and high validity for the assessment of health-related QoL and development in children and adolescents [46]. Increasingly, measures of health-related QoL have been used to evaluate well-being. Health-related QoL conceptualizes “quality of life” as a multidimensional construct which transcends simple physical health outcomes [22,47,48]. PedsQL 4.0 has 23 items, each categorized into various types of functioning, including physical, emotional, social, and school functioning. Examples of PedsQL items include: “It is hard for me to do sports activity or exercise”, “I feel sad or blue”, “I have trouble getting along with other kids”, and “It is hard to pay attention in class”. Students responded to each item by reporting the frequency at which they experienced these occurrences in their daily lives within the month prior to the interview. Respondents could answer each item from “0” for “never” to “4” for “almost always.” For analytical purposes, we followed the PedsQL coding scheme, coding “0” as “100”, “1” as “75”, “2” as “50”, “3” as “25”, and “4” as “0”. We averaged scores to yield a measure of children’s overall quality of life and development. Possible scores ranged from 0 to 100, with higher scores representing greater quality and child development. We administered the Chinese version of PedsQL, which has been assessed for use with children and has been shown to have good reliability (Cronbach alphas between 0.76 and 0.87) [22,47,49]. This study’s Cronbach alpha was 0.90.

The mediating variable of this study was resilience, measured by a concise form of the Resilience Scale, the 14-item Resilience Scale instrument (RS-14) [50,51]. Resilience-related traits were evaluated by the RS-14. It includes personal characteristics that mitigate the destructive influence of adverse life circumstances on proper psychological adjustment [50,51]. Examples of items include: “When I’m in a difficult situation, I can usually find my way out of it”, and “My belief in myself gets me through hard times”. RS-14 shows good cross-ethnic validity in the U.S. and good reliability among Chinese adolescents [52,53]. For this study, the Cronbach’s alpha was 0.91. Participants were instructed to rate each item according to how strongly they identified with each statement when considering themselves over the previous four weeks. Per-item scores ranged from 1 to 7 (strongly disagree to strongly agree). The sum of the item scores was computed and ranged from 14 to 98. Higher scores indicated a higher spot-measurement of resilience.

The independent variables were mindfulness and income. The 14-item Mindful Attention Awareness Scale for Adolescents (MAAS-A) was used to assess mindfulness. The MAAS-A has been validated for adolescents [54] and has good validity and reliability for use of the Chinese version with Chinese adolescents [55]. The 14 items asked participants to identify the frequency over the past four weeks with which they experience feelings, behaviors, or mindful thoughts such as “I rush through activities without being really attentive to them” and “I break or spill things because of carelessness, not paying attention, or thinking of something else.” The score for each item ranged from 1 to 6 (almost never to almost always). We reversed the scores so that higher scores indicated higher levels of mindfulness. The total of all scores provided ranged from 14 to 84, and the Cronbach’s alpha reached 0.88 in this study. Students were asked to report annual family income in the last year.

### 3.3. Analytical Strategy

A descriptive analysis was conducted to assess the distribution of each main variable. Pearson correlation analysis was performed to examine the association between all variables. To further sort out the explanatory power of mindfulness and income on life quality via resilience as a mediator, we conducted Structural Equation Modeling (SEM) to examine the effects of mindfulness and income on both resilience and life quality. SEM differs from ordinary least squares regression techniques in that it allows for not only the examination of direct effects, but also a simultaneous analysis of indirect effects through mediating variables [56]. A path model that depicts the relationships between mindfulness, income, resilience, and life quality is presented in Figure 1. The model posits that mindfulness and income affect life quality of students both directly and indirectly through their respective effects on resilience. Resilience was predicted to directly increase life quality in students. To evaluate model fit, several commonly used fit indices were used, including the comparative fit index (CFI), root-mean-square error of approximation (RMSEA), and the chi-square test. STATA software 16.0 (StataCorp LLC, College Station, TX, USA) was used for all analyses.

## 4. Results

The descriptive statistics and correlation analysis of our main variables are presented in Table 1. Out of 905 students, the mean mindfulness score was 59.4, and scores ranged from 23 to 84. The annual family income in 2019 was, on average, around 23,000 yuan (approximately 3300 U.S. dollars in 2019). The students demonstrated moderate levels of resilience with a mean of 64.2, and sample scores ranged from 19 to 98. Median and mean quality of life score were 78 and 77, respectively, though students’ scores ranged from 31 to 100. The results from Pearson correlation analysis of variables indicated that there was a significant correlation between mindfulness and resilience (r = 0.32, *p* < 0.001) and life quality (r = 0.45, *p* < 0.001). Income was significantly correlated with resilience (r = 0.09, *p* < 0.001), but not with mindfulness (r = 0.01, *p* > 0.05) or life quality (r = 0.01, *p* > 0.04). Resilience was positively correlated with life quality (r = 0.36, *p* < 0.001). Overall, the findings of the correlation analyses were consistent with and supportive of the hypothesized model in Figure 1.

Figure 2 illustrates the results of the fitted model from SEM analysis. The results indicated that there was no direct link between income and life quality. The chi-square value (Figure 2) was 0.2 (*p* > 0.05). The full decomposition of standardized direct and indirect effects of the model is shown in Table 2. The SEM results confirmed that both mindfulness (β = 0.37, *p* < 0.001) and income (β = 0.06, *p* < 0.01) had direct and positive effects on resilience. The reported estimates also demonstrated that both mindfulness (β = 0.41, *p* < 0.001) and resilience (β = 0.23, *p* < 0.001) had positive and direct effects on life quality. Findings from the indirect effect analysis showed that both mindfulness (β = 0.09, *p* < 0.001) and income (β = 0.01, *p* < 0.01) had positive and indirect effects on life quality through their effects on resilience. In sum, the SEM results for this sample demonstrated that mindfulness was strongly and positively associated, through both direct and indirect effects, with life quality. The results showed that income had limited effect on life quality through its effect on resilience. The total effect on mindfulness on life quality was relatively large (β = 0.50, *p* < 0.001), while the total effect of income on life quality was more limited (β = 0.01, *p* < 0.01). Resilience also showed strong effects on life quality (β = 0.23, *p* < 0.001).

In consideration of gender as an important construct in Chinese culture and of the significant difference in life quality scores between male students and female students (79 for males and 74 for females, *p* < 0.001), we further examined the above model by gender, with results listed at the bottom of Table 2. The results based on gender were largely in line with the ones found in the whole sample, except that income had no effect on resilience or life quality for male students, while income showed relatively large effects on resilience and life quality for female students. The reported effects of mindfulness on life quality were large and significant for both male and female students, underscoring the potential positive effects of mindfulness and its concordant effect on resilience as a means to positively increase life quality for high school students.

## 5. Discussion

This paper sought to examine the relationship between family income, an indicator of material capacity, and mindfulness, an indicator of mental capacity, with resilience and quality of life among senior students at a vocational school in China. Our analyses confirmed that both material and mental capacity positively affect quality of life through resilience as a mediating variable. This is consistent with results of several of the studies previously mentioned [12,13,14,20]. Interestingly, however, mental capacity proved to have more of an influence on QoL than that of material capacity—that is, the total effect of mindfulness on QoL was relatively greater than that of income on QoL. To our knowledge, this is one of the first studies that juxtaposes the two, mental capacity and material capacity, with one another to answer the question of whether one has more of an effect on QoL than the other in China.

While Mayer (1997) reviewed several psychosocial factors that explain variability within children’s outcomes, studies focused primarily on parental psychosocial factors. By contrast, our study focused on an individual characteristic, mindfulness, of emerging adults [33]. The results indicated both direct and indirect relationships between income, mindfulness, resilience, and QoL within a sample of Chinese emerging adults. Further, our results showed that money does not necessarily dictate outcomes when considering other aspects. Mindfulness has strong direct and indirect, by way of resilience, effects on QoL than income. This can be explained by the way that mindfulness allows people to effectively cope with otherwise overwhelming thoughts and emotions, without withdrawing or ruminating [15]. Mindful coping breeds resilience, making it possible for individuals to regulate intense emotions, resolve conflicts, and increase QoL.

The finding that mindfulness promotes resilience, which in turn improves QoL, is consistent with past literature confirming the positive relationships between mindfulness, resilience, and well-being [15,57,58]. Studies based in China have also reported on how mindfulness can improve academic functioning, a subsection of QoL [59], as well as buffer against the negative effects of phone addiction on depression and anxiety [60]. Taken together, our findings and those of previous scholars point to mindfulness as an appropriate tool to facilitate emotional regulation among Chinese children and adolescents. Studies have indicated that people of Chinese descent view emotional expression as negative [61] and that Chinese adolescents preferred “downregulation” of their emotions to mitigate intensity and affect [62]. Through mindfulness, individuals may downregulate emotions through attentive, non-judgmental detachment from overwhelming and intense feelings. Beyond this, mental capacity as a stronger determinant of QoL would partially align with studies that have found either no relationship between income and QoL or even a negative relationship, after controlling for several moderators, between the two [63]. Generally, at the macro-level, economic prosperity enhances QoL [64,65], but it is possible that after one’s basic needs are sufficiently met via material capacity, mental capacity serves to encourage day-to-day functioning and therefore improve QoL.

It is also worthy to note that mindfulness continued to have large and significant effects for both male students’ and female students’ resilience and QoL when analyzing our main variable relationships by gender. This is consistent with past findings that mindfulness can be protective against depression, anxiety, and for Chinese adolescents. While income appeared to have no effect on resilience or QoL for male students, data for female students indicated that income had relatively large effects on their resilience and QoL. A possible explanation for this difference may be the gender inequality in intra-household resource allocation within families originating from East and South Asian patrilineal family systems [66]. Asian families have historically shown preferential treatment of sons over daughters [67,68,69]. Kaul’s study of 42,000 households across India found a pro-male bias in educational expenditures, with most expenses going to the eldest son. As for East Asian cultures, a study on intra-household resource allocation in contemporary Japanese society found that resources are allocated not just in favor of sons but away from daughters [68]. Similarly, in China, preference for sons is expressed through the trend of smaller investments in girls’ health [70] and greater spending on sons’ schooling over that of girls [71]. Li and Tsang (2003) found that Chinese parents held higher educational expectations for boys than for girls [71]. It follows, then, that resources will be used to help sons meet those expectations, leaving girls with less. This may explain why income had a significant effect on girls’ resilience and QoL. Girls whose families have higher incomes and can support their education expenses may have greater resilience by way of increased self-esteem [72]. Resource allocation to girls, who are typically cast aside due to preference for sons, may help them feel more valued, thereby increasing self-esteem, then resilience. This is in line with literature that examines the effects of micro-credit and micro-loan programs for women in the Middle East and South Asia. Micro-finance programs have empowered women through increased self-esteem and confidence [73] as well as improved overall well-being [74].

### Limitations

While we present important findings for the future of QoL research, we also acknowledge that our study’s results should be considered in the context of a few limitations. First, analyses were based on a cross-sectional dataset, which can only approximate an associative relationship. Therefore, we cannot conclude a causal relationship between our variables of interest. Future studies can use a longitudinal experimental design to better approximate causal relationships.

In addition to being cross-sectional, all data were collected via student self-reports, leaving room for both intended and unintended reporting errors. For example, students may be inclined to report greater school functioning because they believe that those responses are favored. To counteract this, future studies may choose to triangulate data by collecting responses from multiple sources, including teacher reports and report cards. Next, while the dataset represented a group of emerging adults, it was specific to students in the vocational school system of one province in China. As this was one of few studies to examine income, mindfulness, resilience, and QoL in this specific age group, our results provided preliminary evidence to support the relatively larger role that mindfulness has over income when considering QoL.

Finally, being that the measurements that we used were originally conceptualized, constructed, and tested from the perspective of English-speaking scholars studying English-speaking individuals [75], we must acknowledge the potential fallibility of using such measurements in a sample of Chinese emerging adults, despite the good psychometric properties that were reported. The reliability and validity of translated measurements do not necessitate that such measurements have been appropriately adapted for cross-cultural use [76]. For example, direct translation of idiomatic and colloquial expressions from English to Chinese, though accurate, may not be comprehensible to non-native English-speaking Chinese. Thus, respondents may be unable to accurately relate some of their internal and external experiences according to the scale items.

Beyond assessing reliability and validity, proposed guidelines for cross-cultural adaptations of QoL measures suggest translating, back-translating, committee review of translations and back-translations, pre-testing for cross-cultural equivalence, and re-examining score weighting [75]. Future studies that wish to further examine QoL in Chinese populations, or any non-English-speaking populations, must attend to the shortcomings of translated psychometrics that originate from different cultural contexts.

## 6. Conclusions

Existing scholarly literature has explored several different determinants of QoL; however, a research gap in this area persists. Our study sought to fill the gap in knowledge by examining data from over 900 senior vocational high school students. We found that both income and mindfulness individually had positive relationships with QoL through resilience, though mindfulness had a greater effect on QoL. The results of this study indicated that China’s existing efforts to increase household incomes of families living in poverty can improve emerging adults’ quality of life, albeit in a limited capacity. This points to the need for a more well-rounded approach to improve the quality of life. Additional programs to address mental wellness, possibly through school-based mindfulness interventions, can supplement China’s cash-transfer social welfare programs. School-based mindfulness trainings have shown strong positive effects on the emotional well-being of adolescent students [77,78]. In one study, the mindfulness intervention group reported significant decreases on rumination, intrusive thoughts, and emotional arousal [78]. In another mindfulness-based intervention that incorporated concepts from life skill training, mindfulness was associated with reductions in adolescent emotional and behavioral problems, while life skills were associated with greater resilience [39,55]. Given our findings, interventions such as these can be used to improve the QoL of students. While mindfulness interventions have shown potential to affect participants of any gender, it is equally important that we acknowledge differences in the pathway of income, mindfulness, resilience, and QoL by gender. Income appeared to have a greater effect on QoL for female students than male students. This supports the need for improved outreach to financially support female students, who may be disadvantaged by parents’ preferential treatment of sons. Social programs that seek to explicitly support and empower young females may be of benefit to improve female adolescents’ QoL. Finally, the findings of our study can be expanded upon by replicating our procedures in other schools and by incorporating longitudinal experimental designs. Being that our study was one of the first to explore the difference in effects between income and mindfulness on QoL, as well as their respective pathways, our preliminary findings require further corroboration. Even so, these findings suggest the potential of a supplemental way, through mindfulness, to support adolescent development and improve the QoL of emerging adults.

## Figures and Tables

**Figure 1 ijerph-17-05701-f001:**
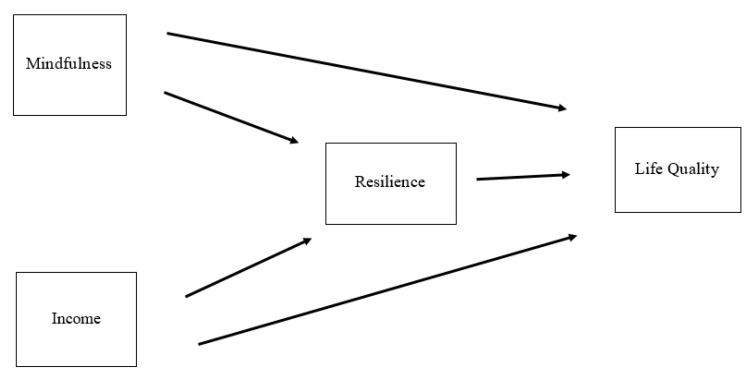
Path diagram of the theoretical model.

**Figure 2 ijerph-17-05701-f002:**
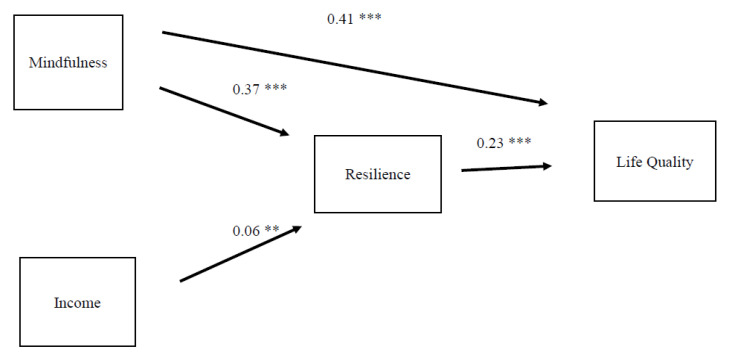
Structural Equation Modeling (SEM) estimates of the fitted model. Chi2(1) = 0.19, *p* = 0.66, standardized root mean squared residual (SRMR) = 0.004; comparative fit index (CFI) = 1.000; root-mean-square error of approximation (RMSEA) = 0.000. *** *p* < 0.001, ** *p* < 0.01.

**Table 1 ijerph-17-05701-t001:** Descriptive statistics and correlation analysis of main variables.

Variable	Mean (S.D.)	1	2	3	4
1. Mindfulness	59.4 (11.0)	---			
2. Income	22676 (20193)	0.01	---		
3. Resilience	64.2 (12.9)	0.32 ***	0.09 **	---	
4. Life Quality	77.0 (12.4)	0.45 ***	0.04	0.36 ***	---

*N* = 905. *** *p* < 0.001, ** *p* < 0.01.

**Table 2 ijerph-17-05701-t002:** Descriptive results of educational benefits (per year) by poverty status.

Whole Sample				
Predictor	Dependent Variable	Direct Effect	Indirect Effect	Total Effect
Resilience	Mindfulness	0.37 ***	---	0.37 ***
	Income ($1000)	0.06 **	---	0.06 **
Life Quality	Resilience	0.23 ***	---	0.23 ***
	Mindfulness	0.41 ***	0.09 ***	0.50 ***
	Income ($1000)	---	0.01 **	0.01 **
*N* = 905				
**Male Sample**				
Predictor	Dependent Variable	Direct Effect	Indirect Effect	Total Effect
Resilience	Mindfulness	0.39 ***	---	0.39 ***
	Income ($1000)	0.03	---	0.03
Life Quality	Resilience	0.22 ***	---	0.22 ***
	Mindfulness	0.34 ***	0.09 ***	0.43 ***
	Income ($1000)	---	0.00	0.00
*N* = 497				
**Female Sample**				
Predictor	Dependent Variable	Direct Effect	Indirect Effect	Total Effect
Resilience	Mindfulness	0.35 ***	---	0.35 ***
	Income ($1000)	0.08 *	---	0.08 *
Life Quality	Resilience	0.18 ***	---	0.18 ***
	Mindfulness	0.53 ***	0.06 ***	0.59 ***
	Income ($1000)	---	0.01 *	0.01 *
*N* = 408				

*** *p* < 0.001, ** *p* < 0.01, * *p* < 0.05.

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
