# Peer review of "Mind Over Matter: Mindfulness, Income, Resilience, and Life Quality of Vocational High School Students in China"

_ijerph, 2020, doi:10.3390/ijerph17165701_

Round 1
Reviewer 1 Report
This is a very interesting, well conceptualized, and literature supported presentation of a research study.
Clarification is sought regarding the following:
- There does not appeared to be any statements or description provided about consent procedures employed with the research subjects to indicate their voluntary participation in the study. Please describe the consent procedures employed to inform subjects about the study.
- Data was collected through both group and self-administered surveys. Please elaborate what each method was and whether there were any differences in the method used that may have influenced the responses from the students participating in the study.
- Please elaborate on the description of "senior vocational students"-are they being prepared for a variety of vocations? specific one (s)? what is the the gender distribution of the sample? Are there specific criteria for entrance into vocational schools in China?
- What is 23 thousand yuan equivalent to in dollars?
- Please provide citation(s) for proposed analytic method for mediation analysis as well as rationale for discarding others (if applicable).
- Cultural traditions, values, beliefs, preferences are suggested as possible explanations for the findings, for example, preference for males over females, is posited as explaining the influence of income. Please comment on the findings from a cultural perspective in regards to mindfulness and resilience.
- Please comment on that although the measures used show good psychometric properties when used in translated forms, they are, never the less, were conceptualized, constructed, tested and measured/scaled from a westernized lens and did not originate from Chinese/ Asian/ Eastern perspective and what are the implications of this for the findings.
Author Response
Reviewer 1:
This is a very interesting, well conceptualized, and literature supported presentation of a research study.
Clarification is sought regarding the following:
- There does not appeared to be any statements or description provided about consent procedures employed with the research subjects to indicate their voluntary participation in the study. Please describe the consent procedures employed to inform subjects about the study.
The following information was added to the Method section: The research protocol was approved by the research review committee at one of the co-authors’ university in China. Informed consent process was implemented prior to the survey. Students were informed of their voluntary participation, as well as the their ability to stop taking the survey at any time.
- Data was collected through both group and self-administered surveys. Please elaborate what each method was and whether there were any differences in the method used that may have influenced the responses from the students participating in the study.
The survey was conducted in self-administered way and in group format (i.e. in the classroom). We revised the text to clearly present the process in the revision.
- Please elaborate on the description of "senior vocational students"-are they being prepared for a variety of vocations? specific one (s)? what is the the gender distribution of the sample? Are there specific criteria for entrance into vocational schools in China?
The secondary vocational education in China is conducted at the high school level and includes a variety of vocational training. School G have 9 majors, including Computing, Accounting, Tourism, Kindergarten Education, Clothing, Machinery, Auto Repair, Electronics, and Aviation. About 55% of students were male. The entrance criterion was based on the nationwide standardized high school entrance exam. Though a maximum score on this exam is 760 points, Guilin Vocational School admits students with scores as low as 280. Meanwhile the criterion for regular high school is a score of 470 or above.
- What is 23 thousand yuan equivalent to in dollars?
23 thousand yuan is equal to approximately $3,300 USD. We have added this into paper to better contextualize the average annual family income of the students.
- Please provide citation(s) for proposed analytic method for mediation analysis as well as rationale for discarding others (if applicable).
SEM differs from ordinary least squares regression techniques in that it allows for not only the examination of direct effects but also simultaneous analysis of indirect effects through mediating variables (Gunzler et al., 2013).
[56] Gunzler D, Chen T, Wu P, Zhang H. Introduction to mediation analysis with structural equation modeling. Shanghai Arch Psychiatry. 2013; 25(6):390–4.
- Cultural traditions, values, beliefs, preferences are suggested as possible explanations for the findings, for example, preference for males over females, is posited as explaining the influence of income. Please comment on the findings from a cultural perspective in regards to mindfulness and resilience.
We have expanded upon the cultural context of the findings, specifically those related to mindfulness and resilience. We situate our findings within those from past studies based in China that have also found similarly positive relationships between mindfulness and academic functioning (Lu et al., 2017). We also report on mindfulness as a protective factor against the effects of phone addiction on depression and anxiety (Yang et al., 2019). Finally, we discuss the use of mindfulness as culturally in-line with reports that Chinese culture implicitly evaluates emotional expression negatively (Deng et al., 2019) and generally prefers “down-regulation” of emotions (Sang et al., 2014).
- Please comment on that although the measures used show good psychometric properties when used in translated forms, they are, never the less, were conceptualized, constructed, tested and measured/scaled from a westernized lens and did not originate from Chinese/ Asian/ Eastern perspective and what are the implications of this for the findings.
We have acknowledged, in the Discussion section (limitations), the implications of using psychometrics that originate from primarily English-speaking scholars for the purposes of studying English-speaking populations. We discuss the guidelines suggested by Guillemin et al. (1993) to appropriately adapt psychometrics cross-culturally, beyond simple translation and reliability/validity measurements.
We thank you for your thoughtful feedback. We have made the appropriate changes to strengthen our paper.
Reviewer 2 Report
The text has interesting assumptions about the quality of life among adolescents (young adults) in China. The data has been collected on a sample of over 900 people. The authors tried to show the relationship between quality of life and variables such as: income; mindfulness; resilience. I have some comments on the text:
- in the summary, it is worth adding information about the research sample and research territory;
- in the introduction there is a reference to HIV patients and (p. 1, line 31). This has no relation to the theoretical framework. It is worth clearing the text from all such references to literature. So remove what is not directly related to the variables under study.
- Point 3.1 is a little unclear to me. In the beginning there were almost 2000 respondents? Finally, 905?
- why did you use a paediatric tool to measure QoL of people in early adulthood?
- the text lacks a procedure to describe the studies, including the ethical layer;
- how was the research sample selected?
- The sociodemographic characteristics of the sample under study are also missing from the text;
- some of the results should be extended with in-depth descriptive QoL statistics (median, gender distribution, metric age, place of residence).
- In part of the discussion you mention that QoL is conditioned more by mental capacity than by economic resources. It is worthwhile to develop and justify this thread more with richer literature.
Recommendations related to mindfulness are in my opinion very useful. The text has valuable assumptions for me, but requires some additions.
Author Response
Reviewer 2:
The text has interesting assumptions about the quality of life among adolescents (young adults) in China. The data has been collected on a sample of over 900 people. The authors tried to show the relationship between quality of life and variables such as: income; mindfulness; resilience. I have some comments on the text:
- in the summary, it is worth adding information about the research sample and research territory;
We have added further detail to the abstract regarding the research sample and research territory, including the sample size and a descriptor to acknowledge that the students attend school in an impoverished county.
- in the introduction there is a reference to HIV patients and (p. 1, line 31). This has no relation to the theoretical framework. It is worth clearing the text from all such references to literature. So remove what is not directly related to the variables under study.
We have removed the specific reference to HIV patients as well as other extraneous references to populations that are not directly related to our study’s sample.
- Point 3.1 is a little unclear to me. In the beginning there were almost 2000 respondents? Finally, 905?
The senior class at the vocational school had 1,004 students, however, 74 students were unavailable for participation in our study. As a result, 930 students participated in the survey. After removing cases with incomplete data, the final analytic sample was 905.
- why did you use a paediatric tool to measure QoL of people in early adulthood?
The Pediatric Quality of Life Inventory is a generic, multidimensional measure of HRQoL in children and adolescents. Given that our study’s participants were in their senior year of high school, the scale fit the age range of the sample.
- the text lacks a procedure to describe the studies, including the ethical layer;
We have added key information regarding ethical standards and consent procedures to indicate voluntary participation in the study on the part of the students.
- how was the research sample selected?
School G is the only vocational school in Gulin County, which has a high poverty rate. Since income is one our key variables, we aimed to understand the effects of income, along with other key variables, on QoL for students in high poverty areas. The selection of Gulin County was based on availability.
- The sociodemographic characteristics of the sample under study are also missing from the text;
We have expanded upon the sociodemographic characteristics of the sample, including gender, age, and ethnicity. About 55% of students were male. The average age of the sample was 18 and 96% of the sample were Han Chinese.
- some of the results should be extended with in-depth descriptive QoL statistics (median, gender distribution, metric age, place of residence).
The median quality of life score was 78. We have added the information into revision as suggested. Age and gender information were added. All students were from Gulin County and nearby villages. Since gender was a key variable in this study, we presented quality of life scores by gender (79 for males and 74 for females, p < .001) in the revision as suggested.
- In part of the discussion you mention that QoL is conditioned more by mental capacity than by economic resources. It is worthwhile to develop and justify this thread more with richer literature.
Recommendations related to mindfulness are in my opinion very useful. The text has valuable assumptions for me, but requires some additions.
We have added more literature on mindfulness to better support the findings. This can be found in our Discussion section. For example, we corroborate our findings with past literature that has already confirmed links among mindfulness, resilience, and quality of life (Bajaj & Pande, 2016; Chamberlain et al., 2016; Hanna & Pidgeon, 2018). Further, we contextualize the material/mental capacity dichotomy with studies that examine QoL in terms of income on an individual level (Tang, 2007) and economic prosperity on a national level (Diener & Diener, 1995; Zagorski et al., 2013).
We thank you for your thoughtful feedback and have carefully incorporated it into our draft to strengthen our paper.
Round 2
Reviewer 2 Report
In the present form, the text meets the conditions for printing. The article outlines the relationship related to: Mindfulness, Income, Resilience and quality of life. The authors in the second version of the text referred to the clarification of issues related to: the research sample, the tool used (PedsQL), the research procedure. Once again, thank you for the opportunity to read the study. I recommend the text for publication. In the article I also find interesting educational threads.